# Phylotypic Diversity of Bacteria Associated with Speleothems of a Silicate Cave in a Guiana Shield Tepui

**DOI:** 10.3390/microorganisms10071395

**Published:** 2022-07-11

**Authors:** Qi Liu, Zichen He, Takeshi Naganuma, Ryosuke Nakai, Luz María Rodríguez, Rafael Carreño, Franco Urbani

**Affiliations:** 1Graduate School of Integrated Science for Life, Hiroshima University, 1-4-4 Kagamiyama, Higashi-Hiroshima 739-8528, Japan; liuqi19950807@outlook.com (Q.L.); he-zichen@hiroshima-u.ac.jp (Z.H.); 2Bioproduction Research Institute, National Institute of Advanced Industrial Science and Technology, 2-17-2-1 Tsukisamu-Higashi, Toyohira, Sapporo 062-8517, Japan; nakai-ryosuke@aist.go.jp; 3Venezuelan Society of Speleology, Apartado 47.334, Caracas 1041-A, Venezuela; luzrodriguezdavila@gmail.com (L.M.R.); arrobaespeleo@gmail.com (R.C.); urbanifranco@gmail.com (F.U.); 4Venezuelan Foundation for Seismological Research, Apartado 76.880, Caracas 1070-A, Venezuela; 5Venezuelan Institute for Scientific Research, San Antonio de los Altos, Miranda 1020-A, Venezuela; 6Department of Geology, Central University of Venezuela, Caracas 1050, Venezuela

**Keywords:** Guiana Highlands, opal-A, bacterial diversity, OTU, microbiome

## Abstract

The diversity of microorganisms associated with speleological sources has mainly been studied in limestone caves, while studies in silicate caves are still under development. Here, we profiled the microbial diversity of opal speleothems from a silicate cave in Guiana Highlands. Bulk DNAs were extracted from three speleothems of two types, i.e., one soft whitish mushroom-like speleothem and two hard blackish coral-like speleothems. The extracted DNAs were amplified for sequencing the V3–V4 region of the bacterial 16S rRNA gene by MiSeq. A total of 210,309 valid reads were obtained and clustered into 3184 phylotypes or operational taxonomic units (OTUs). The OTUs from the soft whitish speleothem were mostly affiliated with *Acidobacteriota*, *Pseudomonadota* (formerly, *Proteobacteria*), and *Chloroflexota,* with the OTUs ascribed to *Nitrospirota* being found specifically in this speleothem. The OTUs from the hard blackish speleothems were similar to each other and were mostly affiliated with *Pseudomonadota*, *Acidobacteriota,* and *Actinomycetota* (formerly, *Actinobacteria*). These OTU compositions were generally consistent with those reported for limestone and silicate caves. The OTUs were further used to infer metabolic features by using the PICRUSt bioinformatic tool, and membrane transport and amino acid metabolism were noticeably featured. These and other featured metabolisms may influence the pH microenvironment and, consequently, the formation, weathering, and re-deposition of silicate speleothems.

## 1. Introduction

Bacterial communities exist in every ecosystem on Earth, and the species compositions of the communities can be adjusted to adapt to various environmental conditions, including caves, as shown by culture-independent pyrosequencing [1]. Bacterial diversity and metabolic strategies in cave ecosystems can improve the comprehension of the biodiversity of different ecosystems around the world, as demonstrated by a metagenomic approach [2]. Although high microbial diversity in caves has been revealed by, for example, clone-by-clone sequencing [3,4], cave microbial communities are still among the least studied [5]. Cave microbial communities are affected by rock types and surface-soil richness/poorness, resulting in various geochemical and hydrochemical features of cave streams, such as pH, organic/inorganic nutrient availability, and buffering action [6,7]. On the other hand, cave microbial communities are likely involved in the formation of various geological forms, such as stalactites in limestone caves [8] and “champignons”, i.e., mushroom-like white speleothems, uniquely found in silicate caves [9]. The roles of microorganisms have been well studied but perhaps are still poorly understood [10].

Silicate caves have been discovered in 1972 in the table mountains, or tepuis, in Guiana Highlands in South America, especially in southeastern Venezuela and western Guiana [11,12,13,14,15,16,17]. Tepuis are table mountains or mesetas composed of Paleoproterozoic quartzites and sandstones, surrounded by steep cliffs [18]. In the process of exploring these isolated environments, karst structures with numerous cave systems and unique silica deposits have been found [15,19]. Due to long-term weathering and water erosion, a variety of silicate speleothems have been formed inside tepui caves. At present, three sets of the world’s largest cave systems in quartzites have been found: one of them in the most popular Roraima-tepui [11,20]; another in the Auyán-tepui, which is known for the world’s tallest waterfall, i.e., 979 m high Angel Fall or *Kerepakupai Merú* [21]; and the third in the Churi-tepui in the Chimantá massif, Venezuela [14]. Other complex networks of underground passages have been found but mainly with deep vertical development; in this case, it is usual to note the extreme scarcity of big speleothems.

In the Roraima-tepui cave system, the microbial involvement in the dissolution of quartz has been implied [15,22]. In another cave in the Roraima-tepui, the silicate cave microflora has been suggested to be affected by limited nitrogen and the poor buffering action of sandstones compared with surface-soil-rich carbonate caves [23]. A microbiological study performed in a cave in the Auyán-tepui showed that both quartz weathering and silica mobility were affected by chemotrophic bacterial communities [24].

This study provides the profiles of MiSeq-generated bacterial phylotypes, or operational taxonomic units (OTUs), based on partial 16S rRNA gene sequences (V3–V4 region) associated with three speleothem samples, two of which were closely similar (i.e., two speleothem types), from the silicate cave system in the Churi-tepui. The compositions and diversity of the retrieved OTUs were generally consistent with those reported for other silicate and limestone caves. The candidate metabolisms of the speleothem-associated bacteria were predicted by the PICRUSt2 bioinformatic tool [25], resulting in the metabolic implication for possible microbial involvement in silicate speleogenesis.

## 2. Materials and Methods

### 2.1. Speleothem Sample Collection

The sampling site was located in a silicate cave of table mountain “Churi-tepui” (ca. 05°15′ N, 62°00′ W) in Chimantá Massif, Gran Sabana, Bolívar State, Venezuela. Chimantá Massif is a dissected plateau with 11 table mountains or tepuis, according to the Pemón Indians’ name, 8 and 3 of which are located in the northern and southern areas, respectively. Churi-tepui belongs to the southern group, having maximum elevation of ca. 2500 m [26] or 2420 m [27] with a summit area of ca. 47.5 km^2^ [26]. Churi-tepui is known for a cave system that accumulates more than 20 km of passages genetically related but currently disconnected in separate caves by breakdowns [14,27]. Charles Brewer Cave is the southernmost of the system and has been mostly studied for geology, geochemistry, and hydrochemistry [15,27], and the pH values of 2.47–2.54 of the acidic cave water was reported for dripping water and “underground river” [27] (p. 38, Table 2). We sampled speleothems formed on the wall of the cave during the Japan–Venezuela Joint Expedition in October 2016.

The speleothems samples were collected at sites on the right bank of the stream between two waterfalls, which were located ca. 540 m from *piedra del helicóptero* at the only cave entrance by straight-line distance (Figure 1). Three speleothem samples were collected from an area of about 5 m^2^ in range and coded as GM1, GM2, and GM3, and their appearances can be roughly seen in Figure 2. GM1 was characterized as whitish and soft-textured (fragile), while GM2 and GM3 were blackish and hard-textured. These speleothem samples were collected by using a flame-sterilized field knife and stored in situ in pre-sterilized Whirl-Pak^®^ plastic bags. The collected samples were then stored below 4 °C for subsequent laboratory experiments.

### 2.2. Crystallographic and Geochemical Analyses

Two speleothem samples, GM1 and GM2, were ground into powder using pre-autoclaved mortar and pestle for crystallographic and geochemical analyses, as well as for the DNA extraction mentioned below; GM3 was wholly expensed for DNA extraction to ensure the recovery of maximally extractable DNA. Although GM3 was not used for the analyses, it was located nearby horizontally to GM2 and resembled GM2 in its blackish appearance and hard texture. The ground powders were analyzed by: energy-dispersive X-ray spectroscopy (EDS) at 10 kV and 15 kV for soft whitish and hard blackish speleothems, respectively; and powder X-ray diffraction (PXRD) by Cu Kα radiation (1.54059 Å) for crystallographic characterization. EDS and PXRD were conducted using JED-2300T (JEOL Ltd., Tokyo, Japan) and RINT2500 (Rigaku Corp., Tokyo, Japan), respectively, at Natural Science Center for Basic Research and Development (N-BARD) of Hiroshima University. In addition, an elemental analysis on the cut surfaces of the speleothems was performed with an electron probe micro-analyzer (EPMA; JXA-iSP100; JEOL Ltd., Tokyo, Japan) at 15 kV at N-BARD. For the EPMA observation, small intact (not powdered) speleothems were embedded in epoxy resin, cut, and polished mechanically at Thin Section Workshop, Craft Plaza of Hiroshima University.

### 2.3. DNA Extraction, PCR Amplification, and MiSeq Sequencing

Bulk DNAs were extracted from approximately 10 g of the ground-powdered GM1, GM2, and GM3 samples with the ISOSPIN Soil DNA extraction kit (Nippon Gene Co. Ltd., Tokyo, Japan) and precipitated in 70% ethanol with precipitation facilitator Ethachinmate (Nippon Gene, Tokyo, Japan). The DNA precipitate was resuspended in sterilized ultrapure water. The concentration and purity of the extracted DNAs were checked with NanoDrop (Thermo Fisher Scientific, Waltham, MA, USA) for subsequent procedures and stored at −20 °C. PCR amplicons were generated using the Kapa HiFi HotStart ReadyMix PCR kit (Kapa Biosystems, Wilmington, MA, USA) and the bacterial V3–V4 region-specific primer pair (S-D-Bact-0341-b-S-17, 5′-CCTACGGGNGGCWGCAG-3′/S-D-Bact-0785-a-A-21, 5′-GACTACHVGGGTATCTAATCC-3′) [28]. The PCR conditions were as follows: 95 °C for 3 min with the lid being heated to 110 °C; 25 cycles of 95 °C for 30 s, 55 °C for 30 s, and 72 °C for 30 s; and a final elongation at 72 °C for 5 min. The sequence library was constructed following our previous method [29]. Pair-end 300 bp sequencing by MiSeq (Illumina, San Diego, CA, USA) was performed using a Nextera XT Index Kit (Illumina) at Department of Biomedical Science, N-BARD, Hiroshima University.

### 2.4. Statistical and Bioinformatic Analyses of the MiSeq-Generated V3–V4 Sequences

Raw sequence data, or raw reads, generated by MiSeq were processed with the Microbiome Taxonomic Profiling (MTP) pipeline by EzBioCloud (https://www.ezbiocloud.net/contents/16smtp) [28]. Briefly, the merging of the pair-end reads as well as PCR primer trimming were conducted using the EzBioCloud in-house pipeline; in this step, unmerged reads as well as short (<100 bp) or low-quality (averaged Q value < 25) reads were omitted. For quality-checked reads, the identical sequences were de-replicated; then, the non-redundant reads obtained were compared to EzBioCloud 16S rRNA gene sequence database PKSSU4.0, with the option of the target taxon of “bacteria”. Note that, in this curated database, the uncultured taxonomic group is tentatively given by the hierarchical name assigned to the DDBJ/ENA/GenBank sequence accession number with the following suffixes: “_s” (for species), “_g” (genus), “_f” (family), “_o” (order), “_c” (class), and “_p” (phylum). The taxonomic assignment of the reads was performed based on the following sequence similarity cut-offs: species (≥97%), genus (>97% > x ≥ 94.5%), family (> 94.5% > x ≥ 86.5%), order (>86.5% > x ≥ 82%), class (>82% > x ≥ 78.5%), and phylum (>78.5% > x ≥ 75%), where x corresponds to the sequence identity with sequences in the database. Note that these cut-offs were taken from previous studies [30,31] and are default parameters of the MTP pipeline. The reads below those cut-offs at the species or higher levels were appended with the suffix “_uc” (for unclassified). Next, all reads that could not be identified at the species level (<97% similarity) were subjected to chimera sequence detection through comparison with the EzBioCloud chimera-free reference database (https://help.ezbiocloud.net/mtp-pipeline/), and the chimera reads identified were discarded; unmatched and eukaryotic plastid reads were also excluded. The resulting final dataset was used to pick phylotypes, or operational taxonomic units (OTUs), based on the 97% similarity cut-off value. Using the EzBioCloud MTP pipeline, the rarefaction curve was computed and visualized; the alpha-diversity indices, i.e., Shannon, Simpson, and Chao1 indices, were calculated to estimate the evenness/richness of the OTUs of each speleothem.

The beta diversity showing principal component analysis (PCA) and hierarchical clustering based on the UniFrac distance matrix was also calculated to compare OTU compositions among the speleothem samples. Venn diagrams at the levels from phylum to species were also drawn. Biomarker OTUs that discriminated the speleothem microbiomes were specified by the linear discriminant analysis (LDA) [32] and LDA-Effect Size algorithm (LEfSe; http://huttenhower.sph.harvard.edu/galaxy/) [33]. The threshold on the logarithmic LDA score for discriminative features is generally set to 2 and was set to 2.5 in a rock varnish study [34]; this study set it to 4 to only focus on biomarkers with large statistical differences between samples.

Major OTUs were further projected on known human metabolic pathways available at Kyoto Encyclopedia of Genes and Genomes (KEGG; http://www.genome.jp/kegg/, accessed on 18 January 2022) [35] and Phylogenetic Investigation of Communities by Reconstruction of Unobserved States 2.0 (PICRUSt2; https://huttenhower.sph.harvard.edu/galaxy/ accessed on 18 January 2022) [36]. The above-mentioned bioinformatic analyses were also performed using the OmicStudio online tools at https://www.omicstudio.cn/tool accessed on 18 January 2022.

### 2.5. Sequence Data Deposition

The raw sequence data, project data, and sample data were deposited in DDBJ Sequence Read Archive (DRA013674), BioProject (PRJDB13191), and BioSample (SAMD00446319 for GM1, SAMD00446320 for GM2, and SAMD00446321 for GM3), respectively.

## 3. Results

### 3.1. Crystallographic and Geochemical Characteristics of Speleothems

The EDS spectra (Figure 3) showed that the major elements of the two speleothem samples were silicon, oxygen, and carbon, clearly indicating that the speleothems were silicate rather than carbonate. Carbon was also detected but as a minor element. The relative abundances of Si and O were different between GM1 and GM2, but the reason was unclear.

The sharp peaks in the PXRD patterns (Figure 4) corresponded to those of quartz according to Powder Diffraction File 00-046-1045 (quartz) of International Center for Diffraction Data (https://www.icdd.com). The difference in the quartz peak intensity between the two samples was due to a greater or lesser inclusion of quartz grains. The lump with a maximum at 22 degrees was the most important feature; it is typical of opal-A [37] and was observed in Venezuelan tepui cave speleothems [27] (p. 84, Figure 79A).

The EPMA images showed interesting contrasts between GM1 and GM2 (Figure 5). The elemental co-occurrence of oxygen and aluminum rather than silicon was seen in GM1, while the co-occurrence of sulfur and calcium was only seen in GM2, but they were very minor and did not affect the overall SiO_2_-dominant features of the speleothems.

Combining the results of the FE-SEM/EDS, PXRD, and EPMA allowed us to identify samples GM1 and GM2 as opal-A, with traces of quartz as impurities from quartz bedrock. The Al element present in sample GM1 probably belonged to pyrophyllite, a metamorphic mineral that has been extensively found in tepui bedrock. Sample GM2 also showed S and Ca, which fits with gypsum also reported for other tepui caves [27,38,39,40].

### 3.2. Evaluation of MiSeq-Generated V3–V4 Sequences (Reads) and OTUs

An overall total of 225,321 sequences, or reads, were generated from the three speleothem samples, 210,309 (93.3%) of which were validated for quality with an average length of 454.0 bp; 15,012 non-validated reads (6.7%) were filtered out due to low quality. For each speleothem, the number and average length of valid reads for GM1 were 87,168 (92.5% of total) and 454.5 bp; for GM2, 49,279 (99.0%) and 458.4 bp; and for GM3, 73,862 (90.8%) and 452.0 bp, respectively (Table 1).

Based on the cutoff at 97% similarity, the valid reads of GM1, GM2, and GM3 were grouped into 474, 1576, and 1134 phylotypes, or operational taxonomic units (OTUs), respectively, for a total of 3184 OTUs. The OTUs were annotated to a total of 1122 bacterial species, 516 genera, 262 families, 142 orders, 76 classes, and 30 phyla, and the hierarchical composition of each speleothem is summarized in Table 1.

The rarefaction curves for reads–OTUs relationships were drawn to calculate the coverage of retrieved OTUs over the predicted total OTUs (=the Chao1 values mentioned below). The coverages for GM1, GM2, and GM3 were 99.33%, 96.66%, and 99.10%, respectively, which showed that the sequencing depth reached in this study was sufficient to describe and characterize microbiomes in the silicate cave speleothems (Figure 6).

The distribution of OTUs among the samples was visualized using a Venn diagram (Figure 7) and the associated table (Table 2) that summarizes the numbers and names of phyla in each intersection and relative compliment (Venn diagrams at the levels from species to class are shown in Appendix A). The intersection of GM1, GM2, and GM3 (GM1 ∩ GM2 ∩ GM3) had the largest numbers of phyla and reads, which showed the relative similarity of the phylum compositions among the speleothems. Eight phyla were specific to GM1 (GM1/GM2/GM3), while GM2 and GM3 had only one specific phylum.

### 3.3. Alpha and Beta Diversity Analyses

The Chao1, Shannon, and Simpson indices were calculated with EzBioCloud to estimate the alpha diversity (within each sample) of GM1, GM2, and GM3 (Table 3). The Chao1 index, an estimator of species richness, corresponding to the predicted OTU numbers in the rarefaction curve analysis (Figure 6), ranged from 490.40 in GM2 to 1586.63 in GM1. The Chao1 index values were almost equivalent to the number of retrieved valid OTUs, which agrees with the “high coverages” of 96.66–99.33% in the rarefaction curve analysis, as “coverage” is defined as the ratio of the number of valid reads to the Chao1 index value of the corresponding sample.

The Shannon index is an estimator of species evenness or diversity of equally abundant species. The highest Shannon index value of 5.19 was calculated for GM1, which was converted to an effective number of species (ENS) of 179.47 by *e*^5.19^ (=2.718^5.19^) [41], that is, the equivalent diversity with 179 equally common OTUs, corresponding to 11% of retrieved OTUs of GM1. Applying this approach to the lower Shannon index values of 3.06 for GM2 and 4.05 for GM3, ENS values as low as 21.33 and 57.40 were calculated, respectively.

The Simpson index is also an estimator of both species richness and evenness, with lower values for higher diversity. As the reciprocal of the Simpson index is regarded as a different expression of the ENS, the highest Simpson index (lowest diversity) of 0.16 for GM2 was converted to an ENS of 6.25, and the lowest Simpson index (highest diversity) of 0.02 in GM1 yielded an ENS of 50 [37].

Beta diversity, i.e., similarity/dissimilarity among samples, was represented by PCA and hierarchical cluster analysis. The PCA showed that the OTU populations of the GM1, GM2, and GM3 speleothems were distinguishable (Figure 8, left). The hierarchical cluster analysis showed a closer similarity between the GM2 and GM3 OTUs, rather than similarities with GM1 (Figure 8, right).

The LEfSe analysis identified the biomarker OTUs that affected PCA-based grouping and hierarchical clustering and discriminated the speleothem microbiomes. The cladogram generated by LEfSe showed that the discriminative biomarker OTUs were affiliated with taxa at various ranks (Figure 9). For example, OTUs affiliated with candidate phylum AD3 or “*Candidatus* Dormibacteraeota” (code “f” in the cladogram) were biomarkers of the GM1 microbiome.

LEfSe identified a total of 100 biomarkers with LDS scores >4 (*p* = 0.01183), of which 14 biomarkers had LDA scores >5, i.e., log_10_5 or 10^5^ (Table 4). The GM1 microbiome was characterized by the OTUs affiliated with acidobacterial class *Solibacteres*, while the GM2 and GM3 microbiomes were characterized by the OTUs affiliated with gammaproteobacterial genus *Dyella*. GM3 was also partly influenced by the OTUs affiliated with *Solibacteres*.

### 3.4. Microbiome Taxonomic Compositions

An overall total of 30 bacterial phyla were identified, of which 28, 20, and 19 phyla were found in the GM1, GM2, and GM3 speleothems, respectively (Table 1).

The dominant bacterial phyla in GM1 were *Acidobacteriota* (37.48% of total OTUs in GM1), *Pseudomonadota* (33.47%), and *Chloroflexota* (9.69%); in GM2 they were *Pseudomonadota* (73.52%), *Acidobacteriota* (12.37%), and *Actinomycetota* (10.10%); and in GM3, *Pseudomonadota* (44.58%), *Acidobacteriota* (35.09%), and *Actinomycetota* (9.86%) (Figure 10). The OTUs attributed to phyla *Pseudomonadota* and *Acidobacteriota* accounted for 70.95%, 85.89%, and 79.67% of the GM1, GM2, and GM3 OTUs, respectively; thus, the predominance of these phyla was revealed in the studied silicate cave speleothems.

*Pseudomonadota*, *Acidobacteriota,* and *Actinomycetota* represented >70% of the studied microbiomes and as high as 95.99% of the GM1 microbiome. However, the relative abundance among samples was different (Figure 10). The other phylum present in all samples was *Chloroflexota*, which accounted for more than 5% in the GM1 and GM3 samples. For other low-abundance phyla, the proportion of the *Nitrospirota* phylum detected in the GM2 sample was greater than 5%, and the proportion of *Pl**anctomycetota* phylum detected in the GM3 sample was greater than 2%.

### 3.5. Microbial Metabolic Functions Predicted by PICRUSt2

All OTUs of each speleothem were used for the PICRUSt2 analysis to characterize the functional metabolic profile of each microbiome. At Level 2, which is a high hierarchical level of the KEGG metabolism category classification, each speleothem microbiome had the same 39 pathways (Figure 11). The “membrane transport” pathway accounted for the highest relative abundance in GM1, while “amino acid metabolism” was the highest in GM2 and GM3 and the second highest in GM1. In both pathways, GM1 showed higher relative abundances, while that of “carbohydrate metabolism” was higher in GM2 and GM3.

At KEGG Level 1 (Appendix A), each microbiome had seven large pathways, which were “metabolism”, “genetic information processing”, “unclassified”, “environmental information processing”, “cellular processes”, “human diseases”, and “organismal systems” in the order of relative abundance. The relative abundance of the “metabolism” pathway was close to 50% in all the microbiomes.

At Level 3 (Appendix A), each microbiome had 251 individual pathways. The “transporters” pathway had the highest proportion (4.36%) in GM1 and the second highest proportion in GM2 and GM3. The pathway categorized as “general function prediction only” had the second highest proportion (ca. 3%) in GM1 and the highest proportion (3.85%) in GM2 and GM3. Another transport pathway, “ABC transporters”, was the third most abundant in GM1 (2.83%) but was relatively low in GM2 and GM3. In contrast, GM2 and GM3 had relatively high abundances (ca. 3%) of “bacterial motility proteins”, “secretion systems”, and “two-component system”, which were higher than in GM1.

## 4. Discussion

Guiana Shield tepuis present a unique geographical location, special geomorphological features, and some of the longest and deepest silicate caves in the world [20,42]. The mineralogical processes had a very long timespan to occur. The first date record from tepui silica speleothems revealed ages from 53 to 390 thousand years [43,44], so the scarcity of elements and the harsh conditions are counterweighted by the availability of longer periods for biological and mineralogical interactions. The carbon only insignificantly detected in EDS spectra of the speleothems was probably due to organic materials that are abundant in the soil of Chimanta Massif tepuis [27] (p. 42). The relative abundances of Si and O were different between the two speleothem types (Figure 3), which could be related to the hardness or softness (fragility) of the speleothems in relation to the water of crystallization and the age of speleothems. The quartz grains found in the speleothems (Figure 4) were probably impurities derived from sandstone bedrock [27] (p. 80, Figures 78–80, Table 5). The EPMA revealed the co-occurrences of Al and O in speleothem GM1 as well as Ca and S in GM2 (Figure 5), which may indicate occurrences of pyrophyllite (Al_2_Si_4_O_10_(OH)_2_) and gypsum (CaSO_4_), respectively, in the speleothems [27] (p. 66, Table 4; p. 85, Figure 75).

Only the results of culture-independent, MiSeq-generated V3–V4 sequences were reported in this study, although cultivation using the R2A agar plates was tried and resulted in the Sanger sequencing of 16S rRNA genes of four isolates, which were related to *Bacillus* and *Paenibacillus* spp., as shown in previous studies on limestone cave microflorae [45,46]. However, their V3–V4 sequences were not found in our dataset from Charles Brewer Cave. They might have been cultured with certain bias or as contaminants and thus were not included in this study.

This study focused on MiSeq-generated V3–V4 sequences and found a total of 30 bacterial phyla in the speleothems of a Guiana tepui silicate cave. In general, the representative phyla were mostly similar to those reported in previous studies (Table 5). Phylum *Pseudomonadota* accounted for the largest proportion of OTUs in this cave, followed by *Acidobacteriota* and *Actinomycetota*. *Actinomycetota* is generally the dominant phylum, accounting for 60% of the bacterial community of limestone caves [6]. In other calcareous cave studies, *Pseudomonadota* was identified as the major phylum [47,48,49,50], which is consistent with the result of this study. A study in a Venezuelan orthoquartzite cave showed the dominance of classes *Actinomycetales* and *Alphaproteobacteria* in endolithic bacterial communities close to the cave entrance [23].

Phylum *Chloroflexota* is a ubiquitous phylum and is often found in caves [50,51]. Bacterial communities in a Venezuelan orthoquartzite cave were dominated (82–84%) by class *Ktedonobacterales* of phylum *Chloroflexota* [23], which was the first identification of class *Ktedonobacterales*; we also identified the OTUs of this class, with the read number accounting for >10% of total reads.

The OTUs affiliated with phylum *Nitrospirota* were mainly found in GM1 (Figure 10). *Nitrospirota* occur in different cave systems, such as the extremely acidic Frasassi Cave in Italy [52], as well as the limestone Pajsarjeva jama Cave and Tito Busillo Cave in Spain [3]. In Oylat Cave in a Turkish marble formation, *Nitrospirota* was the fourth abundant phylum [53]. Species of *Nitrospirota* contribute to nitrogen cycling by nitrite oxidation and provide nitrogen sources to oligotrophic lava cave habitats [54].

The OTUs affiliated with photoautotrophic phylum *Cyanobacteria* were found in the speleothem samples. Possible routes of cyanobacterial intrusion are air flow, infiltration with the water flow, and macrobiological vectors such as spiders carrying certain microorganisms and nutrients to underground [39] (p. 34, Photo 3), while entomological resources and the influence of trogloxen fauna such as bat colonies are generally scarce in tepui caves. Silica precipitation and speleothem formation are attributed to the activities of filamentous bacteria, including cyanobacteria [24]. In this study, *Cyanobacteria* comprised as high as 5% of the total OTUs in three samples.

The OTU-based metabolic prediction of the speleothem microbiomes suggested the presence of chemoautotrophic bacteria, which would support the sustenance of complex microbial communities in the studied silicate cave. Natural nutrient inputs and biogeochemical microenvironments together may have a mutual influence between silica amorphization and microbiomic composition [21]. Local changes in pH and the production of metabolites that influence silica solubility can result from bacterial metabolic processes related to chemoautotrophic activities, e.g., CO_2_ fixation and inorganic nitrogen transformation [55,56]. In this respect, GM2 and GM3 showed high abundances of the OTUs affiliated with genus *Rhizobiales* (family *Methylocystaceae*) capable of N_2_ fixation. The OTUs affiliated with nitrite-oxidizing *Nitrospirota*, whose presence might be correlated to CO_2_-fixation-coupled ammonia oxidation [57], were detected in all the samples. Previous studies also indicated members of *Rhizobiales* and *Actinomycetota* to be involved in biomineralization processes and rock weathering in cave environments [58]. In this study, the amount of *Rhizobiales* in the three speleothem samples was as high as 19%. In addition, ammonia-oxidizing archaea, i.e., members of *Thaumarchaeota*, are generally dominant in the microflora of silicate caves [23] and biodegraded limestone walls (not limestone caves) [59]; therefore, archaeal OTUs should be studied with our DNA samples in future studies.

Samples GM2 and GM3 belonged to the same type of speleothems, and the microbiomic functions as predicted by PICRUSt2 showed high similarity. At KEGG Level 1, “metabolism” accounted for the highest proportion of all pathways in all the speleothem samples. Bacterial metabolisms produce corresponding metabolites, and metabolites secreted from cells can affect the near-bacteria microenvironment. At KEGG Level 2, the high proportion of the “membrane transport pathway” was consistent with the secretion of metabolites. At Level 2, “amino acid metabolism” accounted for the highest proportion, and amino acid metabolisms can change the pH of the microenvironment, which may promote the dissolution and re-precipitation of speleothems; the pH values of the studied cave waters as macroenvironments were reported as 2.52 for dripping water and 2.47–2.54 for “underground river” [27] (p. 38, Table 2). Pathways related to “carbon metabolisms” such as glycolysis/gluconeogenesis, citric acid cycle (TCA cycle), pentose phosphate pathway, and pyruvate metabolism were found at KEGG Level 3. The nitrogen metabolism, sulfur metabolism, and carbon fixation pathways were also found at Level 3. These metabolisms can produce a variety of organic and inorganic acids, thereby changing the near-bacterial microenvironment, where discrete condensation may allow elements to stay longer than in flowing or dripping water.

At present, cave microbiology belongs to a new field in biology and geology. A large part of various and numerous cave ecosystems has not been studied, and the diversity and functional potentials of cave microorganisms remain to be explored by both culture-dependent and -independent methods [60]. In our study, while understanding the diversity of bacteria in the silicate cave, we also suggested bacterial functions in rock transformation, particularly the formation/dissolution of speleothems; in addition, we aim to elucidate archaeal functions in future studies.

**Table 5 microorganisms-10-01395-t005:** Comparison of representative phyla reported from silicate caves, a lava tube, limestone caves, and building walls. Sequencing methods (Seq.), target sequences, and numbers of reported phyla of domains *Bacteria* and *Archaea*. Archaeal phyla are underlined. The sequencing method of “Pyro” indicates 454 pyrosequencing. Phylum names are updated according to the latest valid names [61]. Archaeal phyla are underlined.

Source Cave/Site	Seq.	Target	Phylum		Reference
			No.	Representatives	
Silicate cave	MiSeq	V3–V4	30	*Acidobacteriota, Pseudomonadota, Actinomycetota, Chloroflexota, Nitrospirota*	This study
Silicate cave	Sanger	16S rRNA gene	9	*Chloroflexota, Thaumarchaeota, Acidobacteriota, Pseudomonadota, Actinomycetota*	[23]
Silicate cave	MiSeq	V4–V5	17	*Pseudomonadota, Acidobacteriota, Actinomycetota, Planctomycetota, Chloroflexota*	[24]
Lava tube	Pyro	V1–V3	18	*Actinomycetota, Pseudomonadota, Nitrospirota, Acidobacteriota, Bacteroidota*	[54]
Limestone cave	Pyro	V6	33	*Actinomycetota, Pseudomonadota, Acidobacteriota*	[1]
Limestone cave	Pyro	Metagenome	17	*Pseudomonadota, Actinomycetota, Planctomycetota, Thaumarchaeota, Bacillota*	[2]
Limestone cave	Sanger	V3	6	*Pseudomonadota, Acidobacteriota, Actinomycetota, Planctomycetota, Bacteroidota*	[3]
Limestone cave	Sanger	16S rRNA gene	5	*Pseudomonadota, Actinomycetota, Bacteroidota, Chloroflexota*	[6]
Limestone cave	MiSeq	V3–V4	19	*Pseudomonadota, Actinomycetota, Bacillota, Acidobacteriota, Bacteroidota*	[8]
Limestone cave	Pyro	V4	41	*Pseudomonadota, Bacteroidota, Actinomycetota, Bacillota, Verrucomicrobiota*	[44]
Limestone cave	Sanger	16S rRNA gene	6	*Pseudomonadota, Acidobacteriota, Actinomycetota, Planctomycetota, Bacteroidota*	[47]
Limestone cave	Sanger	16S rRNA gene	4	*Pseudomonadota, Actinomycetota, Bacteroidota, Bacillota*	[48]
Limestone cave	Sanger	16S rRNA gene	7	*Pseudomonadota, Actinomycetota, Bacteroidota, Bacillota, Nitrospirota*	[49]
Limestone cave	MiSeq	V4	12	*Pseudomonadota, Acidobacteriota, Bacillota*	[50]
Limestone cave	Sanger	16S rRNA gene	10	*Pseudomonadota, Bacteroidota, Verrucomicrobiota*	[52]
Limestone cave	Pyro	V6	10	*Pseudomonadota, Actinobacterium, Acidobacterium, Bacteroidota, Verrucomicrobiota*	[53]
Limestone cave	Pyro	Metagenome, V4	54	*Pseudomonadota, Thaumarchaeota, Actinomycetota, Planctimycetota, Euryarchaeota*	[56]
Limestone cave	MiSeq	V3	48	*Actinomycetota, Pseudomonadota, Acidobacteriota, Bacillota*	[57]
Building wall	MiSeq	V3–V4	32	*Actinomycetota, Cyanobacteria, Pseudomonadota, Euryarchaeota, Thaumarchaeota*	[59]

## 5. Conclusions

MiSeq V3–V4 microbiomics was performed for the first time on opal speleothems from a silicate cave. The microbiomes of the soft whitish and hard blackish speleothems were separated by differential analysis; however, dominant phyla such as *Acidobacteriota*, *Actinomycetota*, *Chloroflexota,* and *Pseudomonadota* were mostly similar to those reported for limestone caves as well as silicate caves. The occurrence of *Nitrospirota* was specific to the soft whitish speleothem, which may be related to the biogeochemical cycling of nitrogen. The metabolic features of the speleothem-associated microbiomes were inferred based on the V3–V4 sequences, and the inferred membrane transport and amino acid metabolism may have influences on speleological processes.

## Figures and Tables

**Figure 1 microorganisms-10-01395-f001:**
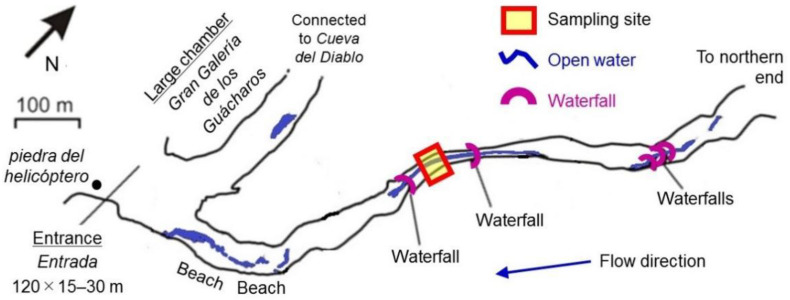
Site of sample collection in Cueva Charles Brewer (Charles Brewer Cave), Churi tepui. The cave map was modified from [9,12].

**Figure 2 microorganisms-10-01395-f002:**
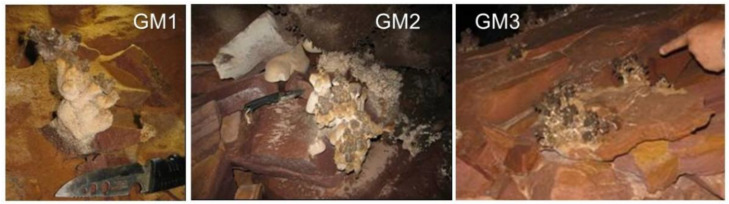
Morphological appearances of the speleothem samples. GM1: speleothem shows a morphology similar to “champignons”. GM2 and GM3: speleothems show coral-like morphology. The lengths of the knife blade and handle in the left and middle photos were 9 cm and 12 cm, respectively.

**Figure 3 microorganisms-10-01395-f003:**
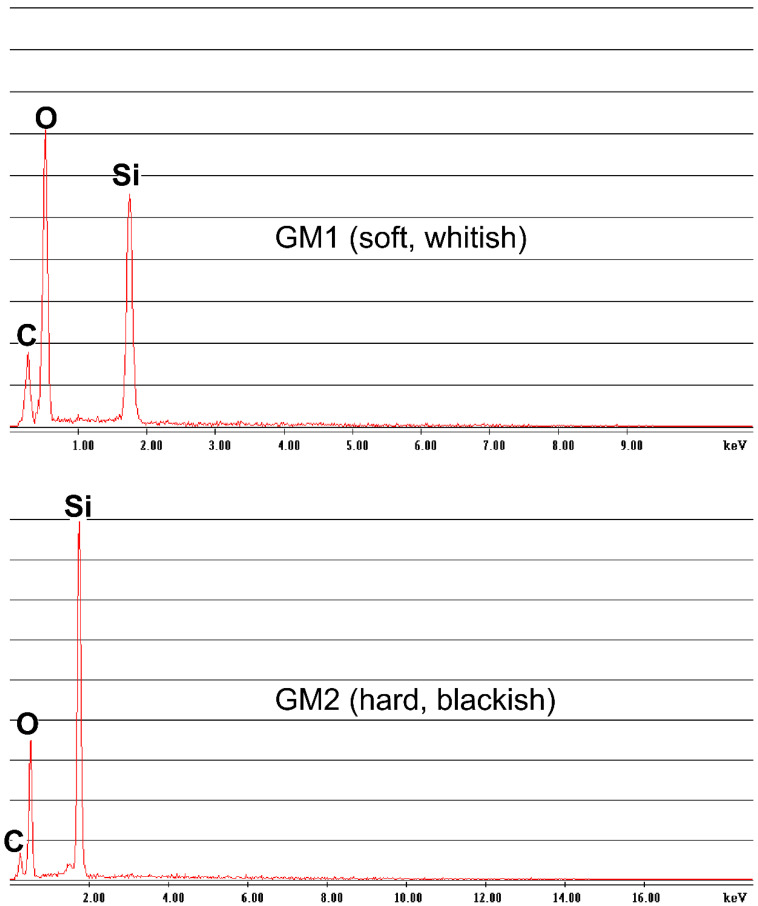
EDS spectra of two speleothem types, GM1 (upper) and GM2 (lower).

**Figure 4 microorganisms-10-01395-f004:**
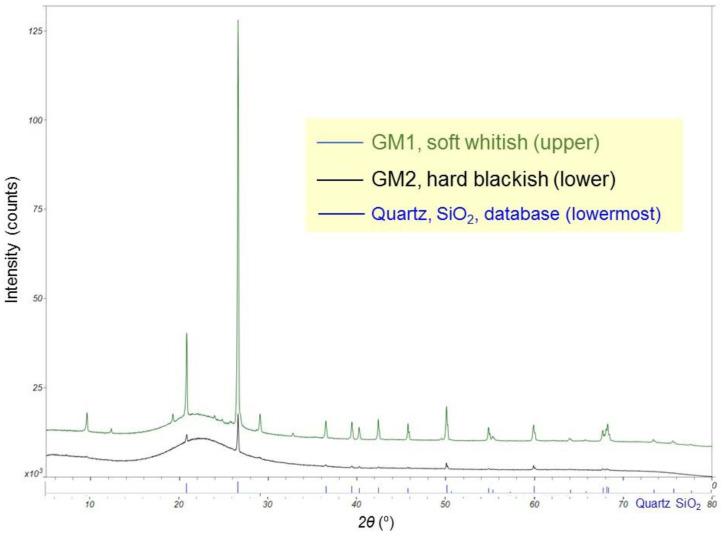
PXRD patterns of two speleothem types, GM1 (upper) and GM2 (lower), with the database pattern of quartz, SiO_2_ (lowermost).

**Figure 5 microorganisms-10-01395-f005:**
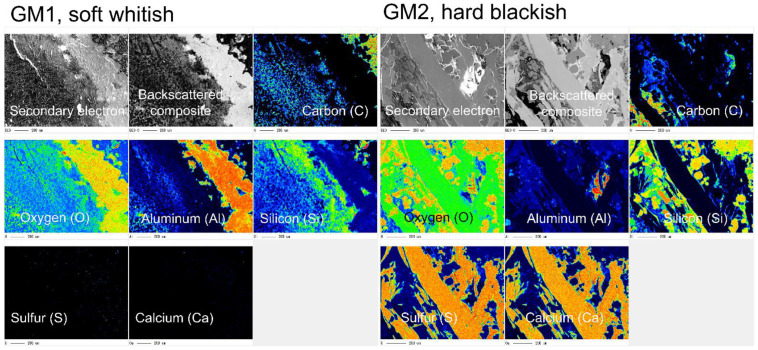
EPMA images of cross-sections of two speleothems, GM1 (**left**) and GM2 (**right**). Warm and cool colors indicate high and low relative abundances, respectively; black areas reveal the absence or less-than-detectable presence of elements.

**Figure 6 microorganisms-10-01395-f006:**
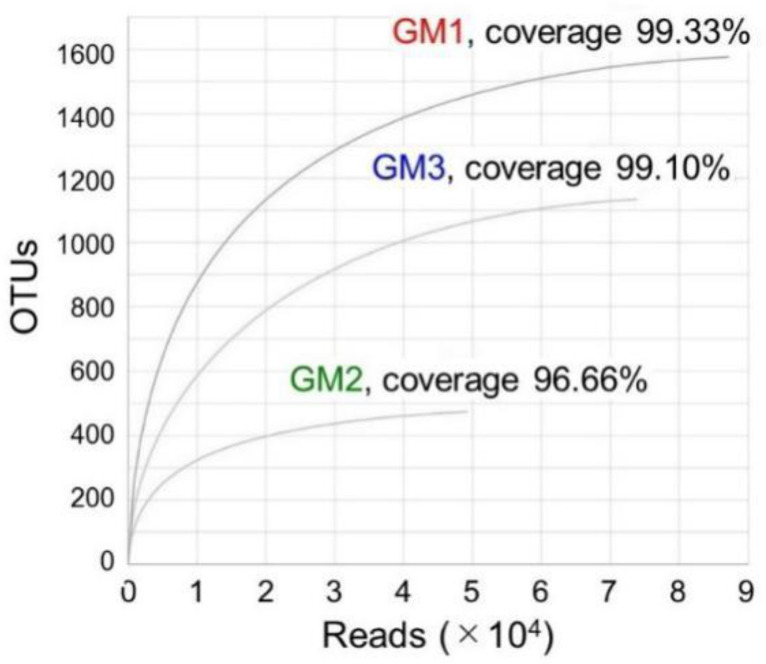
Rarefaction curves based on the numbers of reads and OTUs for the GM1, GM2 and GM3 speleothems.

**Figure 7 microorganisms-10-01395-f007:**
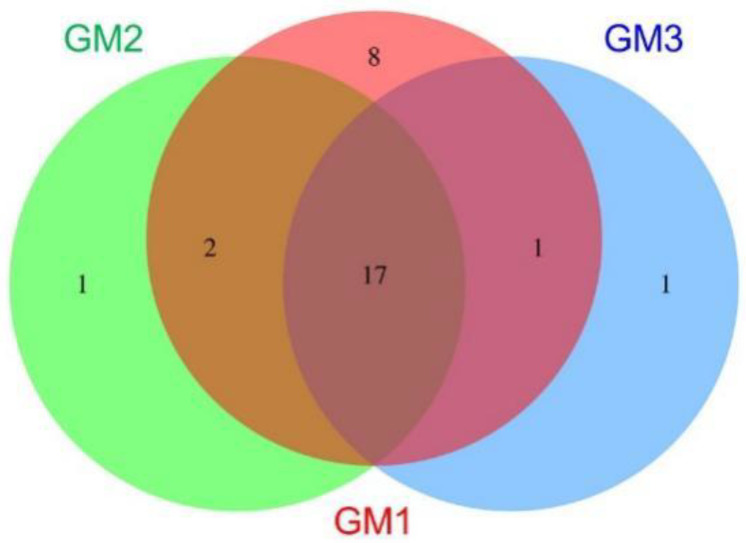
Venn diagram showing the distribution of OTU-affiliated phyla in GM1, GM2, and GM3 speleothems and their intersections. Names of phyla in each intersection and relative compliment are shown in Table 2. Venn diagrams for species, genera, families, orders, and classes are shown in Appendix A.

**Figure 8 microorganisms-10-01395-f008:**
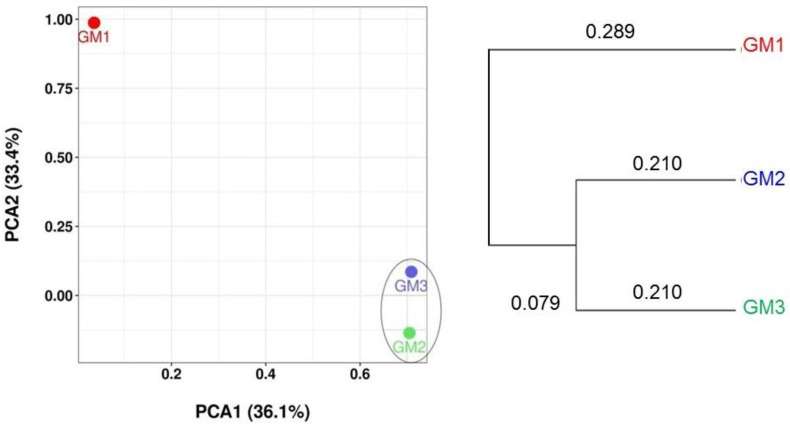
PCA-based grouping (**left**) and hierarchical clustering dendrogram (**right**) of OTU-affiliated species in GM1, GM2, and GM3 speleothems. PCA and dendrogram for genera, families, orders, classes, and phyla are shown in Appendix A.

**Figure 9 microorganisms-10-01395-f009:**
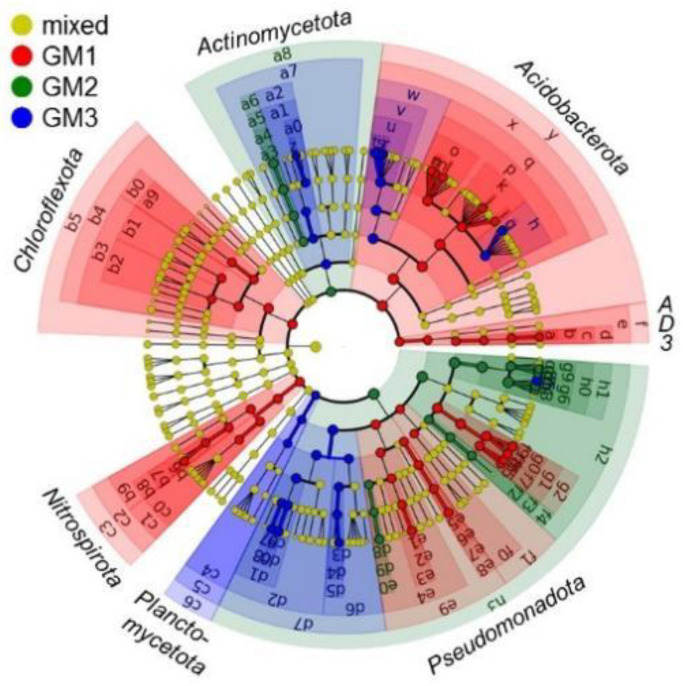
LEfSe cladogram showing taxonomic biomarkers for speleothems GM1, GM2, and GM3. The innermost node corresponds to the Bacteria domain, followed by the concentrically arranged nodes of class, order, family, genus, and species. Red, green, and blue nodes/shades indicate taxa that are significantly higher in relative abundance. The diameter of each node is proportional to the abundance of the taxon. Codes in the cladogram with corresponding taxonomic ranks/names are listed in Appendix A.

**Figure 10 microorganisms-10-01395-f010:**
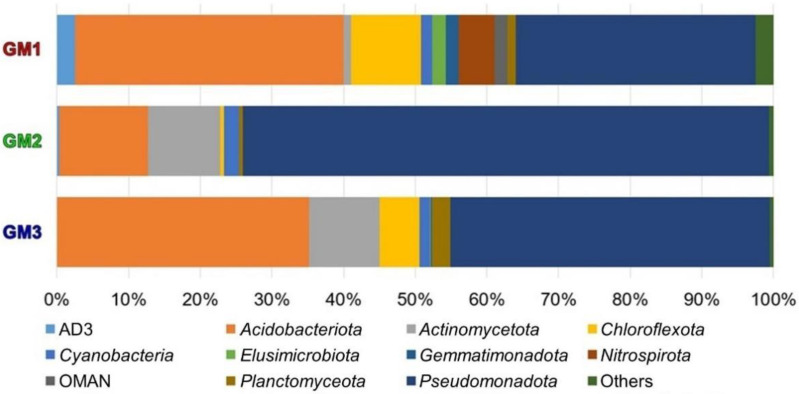
Taxonomic compositions of OTUs from the GM1, GM2, and GM3 speleothems. Nine bacterial phyla and two phylum-level lineages (AD3 and OMAN) were observed with >1% sequence abundance in at least one sample in the MiSeq read data.

**Figure 11 microorganisms-10-01395-f011:**
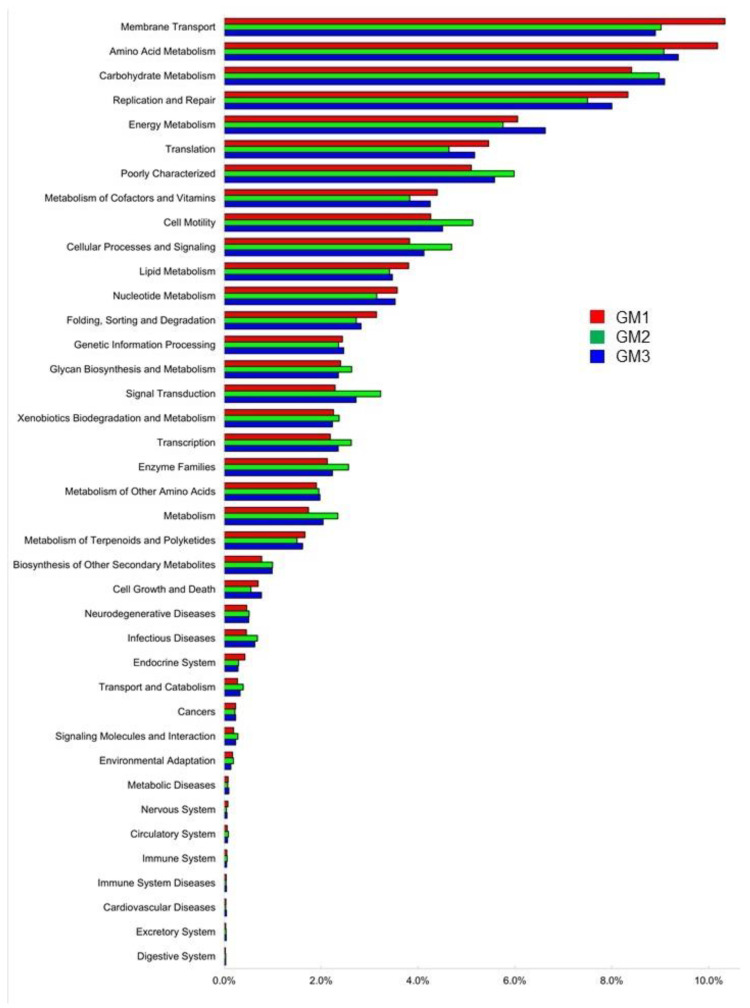
KEGG Level 2 metabolic pathways of GM1 (red), GM2 (green), and GM3 (blue) speleothem microbiomes. Pathway categories are shown in the order of relative abundances in GM1.

**Table 1 microorganisms-10-01395-t001:** Numbers of MiSeq-generated reads, derived OTUs, and annotated taxa in each speleothem sample and the corresponding overall total numbers. Note that the overall total taxa numbers are smaller than simple sums of those for GM1, GM2, and GM3 due to overlaps among samples.

Sample	Raw Read	Valid Read	Phylotype (OTU)	Species	Genus	Family	Order	Class	Phylum
GM1	94,205	87,168	1576	574	326	189	115	67	28
GM2	49,763	49,279	474	298	166	90	61	37	20
GM3	81,353	73,862	1134	365	239	137	83	48	19
Overall	225,321	210,309	3184	1122	516	262	142	76	30

**Table 2 microorganisms-10-01395-t002:** Names of phyla in relative compliments and intersections of the Venn diagram (Figure 7). *Valid names* and candidatus taxa are shown in *italic (oblique)* and roman (upright) styles, respectively, in the order of read abundance.

Speleothem			Phylum	
GM1	GM2	GM3	No.	Name
●			8	Parcubacteria_OD1, Latescibacteria_WS3, Omnitrophica_OP3, Kazan, *Spirochaetota*, Peregrinibacteria, Aminicenantes_OP8, DQ499300_p
	●		1	*Deinococcota*
		●	1	DQ833500_p
●	●		2	*Nitrospirota*, Saccharibacteria_TM7
●		●	1	*Gemmatimonadota*
	●	●	0	
●	●	●	17	*Pseudomonadota*, *Acidobacteriota*, *Actinomycetota*, *Chloroflexota*, *Cyanobacteria*, *Planctomycetota*, AD3, *Elusimicrobiota*, OMAN, *Chlamydiota*, *Bacteroidota*, *Chlorobiota*, *Verrucomicrobiota*, *Armatimonadota*, TM6, *Bacillota*, Microgenomates_OP11
28	20	19	30	

**Table 3 microorganisms-10-01395-t003:** Alpha diversity indices, i.e., Chao1, Shannon, and Simpson indices, for the OTUs for the GM1, GM2, and GM3 speleothems. Effective number of species (ENS) values were calculated from the Shannon and Simpson indices, showing the same tendency of the highest value in GM1 and the lowest value in GM2.

Sample	Valid Read	OTU	Chao1	Shannon (ENS)	Simpson (ENS)
GM1	87,168	1576	1586.63	5.19 (179.47)	0.02 (50)
GM2	49,279	474	490.40	3.06 (21.33)	0.16 (6.25)
GM3	73,862	1134	1144.31	4.05 (57.40)	0.07 (14.29)

**Table 4 microorganisms-10-01395-t004:** Taxonomic biomarkers having LDA scores >5 and their corresponding codes in Figure 9 and Appendix A. All biomarkers with LDA scores >4 are shown in Appendix A.

	Code in Figure 9 and Appendix A	Rank of Biomarker						LDA Score
		Phylum	Class	Order	Family	Genus	Species	
GM1	p	*Acidobacteriota*	*Solibacteres*	PAC000121*_o*	PAC000121*_f*			5.224
	q	*Acidobacteriota*	*Solibacteres*	PAC000121*_o*				5.222
	y	*Acidobacteriota*						5.161
	x	*Acidobacteriota*	*Solibacteres*					5.154
GM2	h0	*Pseudomonadota*	*Gammaproteobacteria*	*Xanthomonadales*	*Xanthomonadaceae*			5.502
	h1	*Pseudomonadota*	*Gammaproteobacteria*	*Xanthomonadales*				5.499
	g6	*Pseudomonadota*	*Gammaproteobacteria*	*Xanthomonadales*	*Xanthomonadaceae*	*Dyella*		5.451
	h2	*Pseudomonadota*	*Gammaproteobacteria*					5.360
	g4	*Pseudomonadota*	*Gammaproteobacteria*	*Xanthomonadales*	*Xanthomonadaceae*	*Dyella*	*D. kyungheensis*	5.283
	h3	*Pseudomonadota*						5.276
GM3	g5	*Pseudomonadota*	*Gammaproteobacteria*	*Xanthomonadales*	*Xanthomonadaceae*	*Dyella*	*D. terrae*	5.077
	v	*Acidobacteriota*	*Solibacteres*	*Solibacterales*	PAC002115*_f*			5.036
	u	*Acidobacteriota*	*Solibacteres*	*Solibacterales*	PAC002115*_f*	PAC002115*_g*		5.021
	w	*Acidobacteriota*	*Solibacteres*	*Solibacterales*				5.011

## Data Availability

The raw sequence data, project data, and sample data are available at DDBJ Sequence Read Archive (DRA013674), BioProject (PRJDB13191), and BioSample (SAMD00446319 for GM1, SAMD00446320 for GM2, and SAMD00446321 for GM3), respectively.

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
