# Peer review of "Phylotypic Diversity of Bacteria Associated with Speleothems of a Silicate Cave in a Guiana Shield Tepui"

_microorganisms, 2022, doi:10.3390/microorganisms10071395_

Round 1

Reviewer 1 Report

The manuscript is titled “ Phylotypic diversity of bacteria associated with speleothems of a silicate cave in a Guiana Shield tepui, Venezuela“. I find the idea interesting and in line with the aim of the journal. I have some concerns about the experimental setup to justify what the authors claim. Moreover, the rationale behind some of the data presented was not entirely clear. I also recommend to the authors improve their references by conducting a more extensive review of international literature. Particularly, the introduction statements are not supported by the references selected by the authors. The logic of some sentences is also questionable. Below are my point-to-point analysis of the manuscript. 

Ø  I suggest modifying the title should be more crisp and brief.

Ø  Abstract introductory statement is too long, it has to be improved with a more specific rationale of the study. The abstract should have crisp information about aim materials method result and conclusion, which I don't find in the present form of an abstract.

Ø  Introduction of the manuscript is too short with fewer references. I recommend adding a write-up to the introduction. 

SStatment "Silicate caves have been discovered since 1972 in the table mountains, or tepuis, in the Guiana Highlands of South America, especially in southeastern Venezuela and west-ern Guiana [7–12], as reviewed in [13]." sentence should not end with a number, I suggest to write scientist group name and then cite at the end.

Materials and methods

  Could you please throw more light on the procedure employed?

Results and discussion:

Ø  Although this section looks okay, I suggest comparing it with more studies of similar nature. 

Ø  Conclusion: This section seems to be missing, I COULD NOT FIND. The author should add this section in the revised MS.I find this paper very interesting and important as it promotes the use of soil organic amendments like biochar hence recommend it is accepted and published after it is fine-tuned.

AAlthough the study is interesting and could be useful for a certain group of the scientific community, therefore, I would suggest improving the manuscript, giving a chance for the next round, because the subject is interesting

Reviewer 2 Report

Comments:

The paper” Phylotypic diversity of bacteria associated with speleothems of a silicate cave in a Guiana Shield tepui, Venezuela” by Liu et al. is interesting and focuses on case study of Geomicrobiology in a type of environment that still demands study efforts, as are the silicate caves, and having knowledge about the communities that thrive in this type of environment is important.

The paper is very clear, very well written and well designed and organised.

However, I recommend to change the title to something like this: “Bacterial structural diversity associated with speleothems of a silicate cave in a Guiana Shield tepui, Venezuela assessed by NGS/ by Illumina” or “Bacterial Phylotypes associated with speleothems of a silicate cave in a Guiana Shield tepui, Venezuela assessed by NGS”. In fact, I do always prefer the designation “OTU -  operational taxonomic units” to “Phylotype”. The word “Phylotypic” is not used very often and I believe that this word is more related to Embryology. Nevertheless, it is just a recommendation as the term is found in a couple of similar papers. Actually, one example is referred in this article.

The abstract is clear, pointing out the main results.

The Introduction is very well written, and the objectives are clearly stated.

Methods are appropriate and are well explained. The used techniques are adequate to fully respond to the aim of the study.

Nevertheless, only 3 samples were considered. Do the authors find that this is enough to have a broad picture? Did the pH and other abiotic conditions and factors vary between these 3 sampling points? For example, authors give values of pH from the literature for the cave water, but we do not know if the pH of the 3 different speleothems are the same…Moreover, it would be interesting to have data on the salinity, temperature and humidity values also, if possible.

Why did not the authors perform the profiling of archaeal diversity? There are groups of Archaea, for example Thaumarchaeota (class Nitrososphaeria), which is a taxonomic group that may have a dominant role in the oxidation of ammonia, being designated as ammonia-oxidizing archaea. Ammonia-oxidizing archaea are ubiquitously detected in many natural environments, where they have an active role in the nitrogen cycle. They also have been detected in the surface of stone building where their presence and activity has been related to events of biodeterioration. Please see for example the work by Coelho et al. (2021): Microorganisms 2021, 9(4), 709; https://doi.org/10.3390/microorganisms9040709

Another point that I would like to mention is that the authors did not perform dependent cultivation methods. For me, this is not crucial for this work, but at least a paragraph should be included in the Discussion on this matter, in my humble opinion.

Results are well and extensively described with the appropriate number of tables and figures and supplementary material, clearly presented and are important to the state of the art in this field of Geomicrobiology. The figures/images are of good quality and elucidative.

Note: In Table 3, try to adjust the names of the taxa in columns into one line only, so that there are no letters transposing for the next line below.

Give space before line 265. Paragraph in line 305 does not have a tab.

Discussion of the results is also complete and addresses the most important questions.

Nevertheless, a paragraph should be included in the Discussion about the absence of dependent cultivation strategies, as I mentioned.

Moreover, I think that also a paragraph (maybe after lines 409-411) on the future possibility to study also the archaeal diversity would be interesting, as I said above these microorganisms can have an important role in the Nitrogen cycle. Please see for example the paper referred above by Coelho et al. (2021) “Bacterial and Archaeal Structural Diversity in Several Biodeterioration Patterns on the Limestone Walls of the Old Cathedral of Coimbra” or others like

Mansch, R.; Bock, E. Biodeterioration of natural stone with special reference to nitrifying bacteria. Biodeterioration 1998, 9, 47–64.

Sand, W.; Bock, E. Biodeterioration of mineral materials by microorganisms—biogenic sulfuric and nitric acid corrosion of concrete and natural stone. Geomicrobiol. J. 1991, 9, 129–138.

Other small issues in Discussion: in line 387 the authors allude to the pH and correctly say that these pH changes may be associated to bacterial metabolic processes related to chemoautotrophic activities. I do agree but the authors did not measure the pH of the geological formations GM1, GM2 and GM3 individually, and I do think that is a gap. The authors only present pH values in an interval picked from the literature.

The conclusions at the end of Discussion should be more highlighted and developed, in my opinion. Moreover, I do think that we can face several ecosystems regarding the cave systems around the planet, being them limestone, silicate or other stone substrate, karst caves, etc., and so we cannot talk about only one cave ecosystem. In this way, I do not agree at all with the expression “…the cave ecosystem…” (in singular), and also with the statement that “…a large part of the cave ecosystem…” has already been explored… I think that many caves remain unexplored, at least, in this way of determining microbiomes composition through NGS in great detail.

The list of references is extensive, well formatted and complete! Note: in reference # 7 the title of the journal is not abbreviated. Please check, if possible, you can find the proper abbreviation of that Slovak journal or bulletin.

Reviewer 3 Report

This paper robustly identifies the bacterial communities associated with speleothems from a silicate cave of the Churi-tepui table mountain. It combines microbiological and mineralogical data. The paper is well-written and the concept is interesting, but my main concern is related to the lack of novelty in your research that advances the knowledge on cave microbiology. It solely describes and poorly discusses the microbial diversity of three samples from Charles Brewer Cave. In addition, mineralogical considerations are slightly presented and discussed, and there are no other analyses that would be of interest to a broader cave scientific community.

Abstract:

Although well written I think that indicating the objectives and hypotheses of the study would focus the readers. I also miss a final statement with a key message of your work.

Materials and Methods:

Lines 101-102: The authors mention that GM3 sample was not used for analyses because it was apparently comparable to GM2. If GM3 was collected, please justify why it was not used for mineralogical analysis. It is well-known that mineral deposits with similar appearance can have different mineralogical compositions.

Section 2.4: 

Cave microbiota was analyzed using the 16S-based microbiome taxonomic profile pipeline of EzBioCloud, which is commonly employed for the microbial taxonomy of human microbiome. In the case of environmental samples collected from unexplored environments, it is highly recommended to use the SILVA database (https://www.arb-silva.de).

Results

Lines 178 – 181: This is speculation and relies on several assumptions, which require further studies on different samples, such as percolating water and bedrock. This would give useful information. Also, considering the significance of this type of speleothems for paleoclimate, geochronological analysis would be of great interest. Hence, overinterpretation of the origin of the mineral deposits should be avoided and statements should be softened. Discussion on mineralogical data should be moved to the Discussion sections. 

Lines 186 – 187: Again, this is speculation, and should be discussed in the Discussion section and supported by literature. 

Lines 196 – 199: Please move it to the Discussion section and extend the discussion on the mineralogical data. 

Lines 232 – 235 and Figure 7: I appreciate Venn diagrams to understand specificities in samples but, in this manuscript, they have been used as a synonym of richness. Instead, I would stress the importance of those specific bacteria (OTUs). Moreover, there is awareness that Illumina Miseq does not confer a great taxonomic resolution at the genus or species levels. In this sense, I would not include these diagrams to genus and species levels.

Lines 316 – 319: Also the heatmap (Figure 11) might be discussed better or remove it from the manuscript, since it did not provide further information. 

Discussion:

The Discussion section is very poor and solely devoted to the microbiological data. 

Given the fact that microbe-mineral interactions can promote constructive (biomineralization) or destructive mineral processes (mineral dissolution), the authors did not provide any hint about these interactions in any point of the Discussion nor if there is any relationship with the changes in the bacterial microbiota and mineralogy.

Also, the discussion of changes in the diversity indices is hidden among the sequencing results, and should be worked out. 

Round 2

Reviewer 1 Report

The author has modified the manuscript as per the suggestion. The manuscript is now ready for publication. I suggest accepting the manuscript without any further changes. 

Reviewer 2 Report

Dear Authors,

Thank you for adressing my questions and doubts. I think all of them were adressed and answered properly. I find that the article was much improved and now this version is prompt for "Microorganisms", in my opinion.

I have no further questions, unless that as the authors recognize it will be useful also to study Archaea, in the future, in these environments of silocate caves. Congratulations for your work and output!